# G-CUT3R: Guided 3D Reconstruction with Camera and Depth Prior Integration

## Abstract

We introduce G-CUT3R, a novel feed-forward approach for guided 3D scene reconstruction that enhances the CUT3R model by integrating prior information. Unlike existing feed-forward methods that rely solely on input images, our method leverages auxiliary data, such as depth, camera calibrations, or camera positions, commonly available in real-world scenarios. We propose a lightweight modification to CUT3R, incorporating a dedicated encoder for each modality to extract features, which are fused with RGB image tokens via zero convolution. This flexible design enables seamless integration of any combination of prior information during inference. Evaluated across multiple benchmarks, including 3D reconstruction and other multi-view tasks, our approach demonstrates significant performance improvements, showing its ability to effectively utilize available priors while maintaining compatibility with varying input modalities.

## 1 Introduction

The pursuit of robust 3D scene reconstruction, as well as the development of versatile models capable of unifying diverse 3D perception tasks, including depth estimation, feature matching, dense reconstruction, and camera localization, is a complex and long-standing challenge in computer vision and computer graphics. Traditional approaches, such as Structure-from-Motion (SfM) and Multi-View Stereo (MVS) Yao et al. (2018), rely on per-scene optimization, which is computationally expensive, slow to converge, and dependent on precisely calibrated datasets, limiting their practicality in real-world scenarios. This has led to the development of feed-forward methods Fan et al. (2017); Wu et al. (2016); Wang et al. (2024b) as a promising alternative. These models leverage large-scale training data and learned priors to achieve orders-of-magnitude faster inference and improved generalization, making them ideal for time-sensitive and scalable applications like real-time robotic perception and interactive 3D asset creation.

Recent advances in feed-forward 3D reconstruction have placed these methods as compelling alternatives to traditional SfM techniques such as COLMAP Schonberger & Frahm (2016), offering enhanced efficiency and robustness in generating 3D scene representations. In particular, DUSt3R Wang et al. (2024b) has pioneered this paradigm by leveraging pairs of RGB images to simultaneously predict point clouds and camera poses, achieving impressive results with minimal input. Building upon this foundation, subsequent works Leroy et al. (2024); Wang et al. (2025b;a) have significantly extended the capabilities of feed-forward approaches. For example, MASt3R Leroy et al. (2024) improves the robustness of 3D reconstruction by grounding predictions in geometric and semantic constraints, improving accuracy in complex scenes. CUT3R Wang et al. (2025b) introduces a recurrent processing mechanism that sequentially refines reconstructions of image sequences, allowing better handling of temporal and spatial coherence in dynamic environments. Meanwhile, VGGT Wang et al. (2025a) advances DUSt3R's framework by adopting a fully multi-view approach, concurrently utilizing all available images to produce more comprehensive and consistent 3D models.

Despite the significant advancements achieved by DUSt3R (Wang et al., 2024b) and its derivatives, feed-forward 3D reconstruction methods typically rely exclusively on RGB images, neglecting additional data sources such as calibrated camera intrinsics, poses, and depth maps from RGB-D or LiDAR sensors, which are commonly available in real-world applications. Effectively incorporating these diverse modalities to enhance the quality of 3D reconstructions remains a critical challenge in computer vision. Recently, Pow3R (Jang et al., 2025) has been proposed to integrate prior infor-

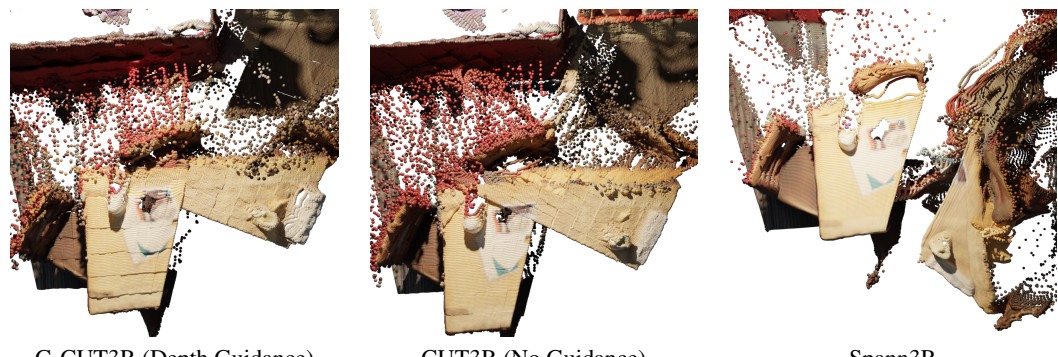

|  G-CUT3R (Depth Guidance) | CUT3R (No Guidance) | Spann3R |

Figure 1: *Comparison of methods.* Visual results across three different approaches: our G-CUT3R with depth guidance, original CUT3R without any guidance, and Spann3R. Our method produces cleaner and more complete 3D reconstruction.

mation into the DUSt3R framework. However, achieving competitive performance requires training the entire model from scratch, which requires substantial computational resources. Furthermore, Pow3R processes only pairs of images and relies on computationally expensive global alignment to produce the final reconstruction, resulting in a runtime of less than 1 FPS. This limitation makes Pow3R impractical for real-time applications.

We propose G-CUT3R, a lightweight and modality-agnostic extension to the CUT3R framework (Wang et al., 2025b) that seamlessly integrates geometric priors through a streamlined encoding process and carefully designed fusion techniques in the decoder stage. Using ray-based encoding for camera parameters and depthmaps, alongside zero-initialized convolutional layers for stable feature integration, G-CUT3R outperforms Pow3R and other state-of-the-art methods. Our approach bridges the gap between traditional SfM techniques and modern feed-forward pipelines, enabling more reliable 3D reconstructions for complex real-time applications. Our carefully designed architecture and training strategy enable G-CUT3R to effectively incorporate any combination of prior information while maintaining robust performance on RGB images alone. Additionally, the recursively updated state eliminates the need for computationally expensive global optimization, making G-CUT3R well-suited for real-time applications. Furthermore, training is performed using only a subset of the original model's parameters and a reduced portion of its training dataset, enhancing efficiency without compromising performance.

Our primary contributions are as follows:

- G-CUT3R, a novel real-time feed-forward method for guided 3D scene reconstruction that utilizes prior information, e.g., camera intrinsics, poses, and depths alongside RGB images.
- Comprehensive experiments demonstrating a significant performance improvement, achieving state-of-the-art results across multiple benchmark datasets and tasks.

## 2  RELATED WORK

**Structure from Motion.** Joint estimation of 3D scene geometry and camera poses from a set of 2D images is a fundamental problem in computer vision Hartley & Zisserman (2003); Crandall et al. (2012); Jiang et al. (2013). Traditional SfM pipelines, such as COLMAP Schonberger & Frahm (2016), rely on a sequence of well-established steps: detecting and matching keypoints across images, estimating initial camera poses and 3D points, and refining these estimates through bundle adjustment. Keypoint matching typically uses hand-crafted features such as SIFT Lowe (1999) or, more recently, learned features such as DIFT Tang et al. (2023). Recent advances have integrated machine learning to enhance various components of the SfM pipeline. Methods like D2-Net Dusmanu et al. (2019), LIFT Yi et al. (2016), and others Chen et al. (2021) employ trained models to improve keypoint detection, descriptor matching, and correspondence estimation, achieving robust performance under challenging conditions. In particular, VGGSfM Wang et al. (2024a) introduced a fully differentiable SfM framework, enabling end-to-end optimization of the reconstruction process.

Although machine learning approaches have significantly improved accuracy and robustness, the core principles of SfM remain rooted in geometric optimization and correspondence-based reconstruction.

**Deep Learning Approaches.** Recent advancements in deep learning have introduced novel alternatives to traditional SfM methods. DUSt3R Wang et al. (2024b) represents a significant deviation from conventional SfM pipelines by predicting point clouds from image pairs without relying on geometric constraints or inductive biases. Unlike traditional SfM, which depends on keypoint matching and geometric optimization, DUSt3R generates predictions in a shared coordinate frame, enabling robust reconstruction across diverse scenes. This approach addresses several challenges inherent in classical methods, such as sensitivity to initialization and sparse correspondences. Building on this paradigm, several works have proposed variations with distinct architectural innovations. MASt3R Leroy et al. (2024) improves the estimation of the pixel-wise correspondence between image pairs, strengthening the efficacy of unconstrained feed-forward models for SfM tasks. CUT3R Wang et al. (2025b) introduces a recurrent formulation of DUSt3R, achieving computational efficiency at the expense of marginal accuracy degradation. More recently, VGGT Wang et al. (2025a) proposes a multi-view architecture that processes multiple images simultaneously, moving beyond pairwise processing to improve reconstruction consistency and robustness.

**Guidance through Prior Information.** While feed-forward deep learning methods, such as DUSt3R, achieve superior results in unconstrained 3D geometry prediction, integrating prior information remains a significant challenge. In many applications, incomplete or noisy geometric priors (such as those derived from LiDAR or similar sensors) are available and can enhance reconstruction accuracy and consistency. Unlike traditional SfM pipelines, which naturally incorporate priors through geometric constraints, fully feed-forward approaches struggle to leverage such information effectively. To address the challenge of integrating prior information, Pow3R Jang et al. (2025) extends the DUSt3R framework by incorporating optional depth and camera pose priors as additional inputs, providing guidance to improve reconstruction quality while maintaining the flexibility of feed-forward models. We extend the CUT3R model with a prior-guided regularization. CUT3R's continuous formulation naturally accommodates informative priors, and its known consistency issues create a clear opportunity for improvement. Our lightweight regularizer boosts both efficiency and accuracy without incurring the memory costs of larger alternatives such as VGGT Wang et al. (2025a).

## 3 METHOD

**Overview.** We introduce G-CUT3R, a novel method that takes as input a set of $\{I_i\}_{i=1}^{N}$ RGB images $I_i \in \mathbb{R}^{3 \times H \times W}$ with the corresponding auxiliary information $\Phi \subseteq \{K, P, D\}$ as guidance to reconstruct the 3D scene. We denote $K \in \mathbb{R}^{3 \times 3}$ as camera intrinsics, $P = [R \,|\, t] \in \mathbb{R}^{4 \times 4}$ as camera pose, and $D \in \mathbb{R}^{H \times W}$ as depth map with corresponding mask $M \in \{0, 1\}^{H \times W}$ for sparse depth. The views are passed sequentially to the network $\mathcal{G}$ and produce 3D pointmaps and camera poses simultaneously, retrieving and updating the state $S$ that encodes the understanding of the 3D scene.

### 3.1 RECAP OF CUT3R

We follow the recently proposed CUT3R method Wang et al. (2025b) that enables efficient processing of a large number of images in a recurrent, memory-constrained manner. CUT3R is presented as a framework that takes a set of images (i.e., either ordered or unordered) as input and outputs corresponding point maps. The process begins with each image $I$ being passed through a Vision Transformer (ViT) encoder, denoted as $E^I$, to extract features $F^I$, defined as $F^I = E^I(I)$. This step leverages the strengths of ViT, which is known for capturing global dependencies in images through self-attention. To maintain context, CUT3R introduces state tokens $s_j$. These tokens interact with the image features via cross-attention in the decoder stage, allowing for mutual updates. The interaction is formalized as: $[z'_j, F'^I_j], s_j = \text{Decoders}([z_j, F^I_j], s_{j-1})$. Here, $z_j$ represents the learnable "pose token" and $[z'_j, F'^I_j]$ denotes the updated features enriched with state information, while $s_j$ is the updated state token. This recursive mechanism ensures that the features of each image are informed by the context of previous images, enhancing the model's ability to understand complex scenes and temporal dynamics.

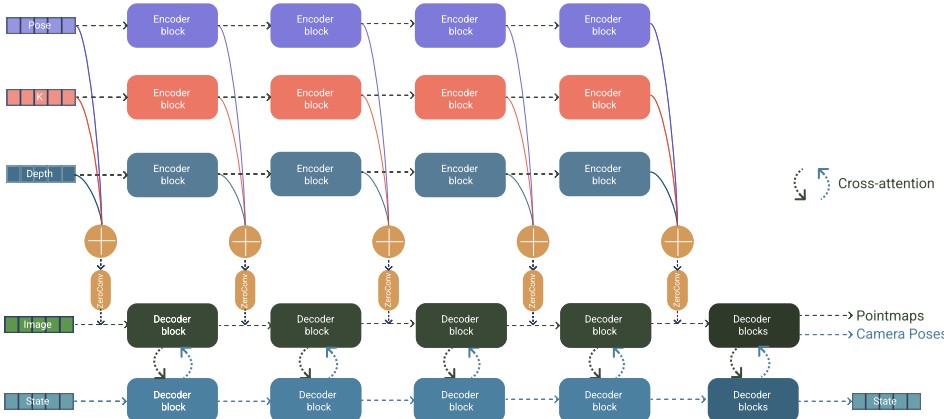

Figure 2: *Overview of the G-CUT3R architecture.* Our method processes a set of RGB images from videos or collections of images, together with a variable set of auxiliary inputs, including depth maps (*Depth*), camera intrinsics ($K$), and camera poses (*Pose*). These inputs are sequentially fed into the network, which employs modality-specific convolutional layers and ViT encoders to extract and fuse features. These features are fused with RGB image features via zero-initialized convolutional layers within the decoder stage, enabling the model to generate accurate 3D pointmaps and camera poses while updating a state token to maintain scene context across sequential inputs.

## 3.2 Incorporating Prior Information

The baseline feed-forward 3D reconstruction pipeline in CUT3R lacks the capability to leverage additional prior information, including camera poses or depth maps, to enhance scene reconstruction accuracy. To address this limitation, we propose a lightweight and modality-agnostic extension to the CUT3R framework, by modifying only the decoder stage to seamlessly integrate additional input modalities, including noisy depth maps and both intrinsic and extrinsic camera parameters, as illustrated in Fig. 2. This approach ensures compatibility with diverse data sources while preserving the integrity of the pre-trained model, making it suitable for advanced tasks such as 3D reconstruction and scene understanding. The flexible design of this extension also holds potential for application to other feed-forward reconstruction pipelines.

**Modality Encoding.** We encode camera intrinsics $K$ and poses $P$ as ray images, representing each pixel $(m, n)$ in an image of resolution $H \times W$ as a normalized 3D direction. This yields $X^K \in \mathbb{R}^{3 \times H \times W}$ and $X^P \in \mathbb{R}^{3 \times H \times W}$, computed as follows:

$$X^K = \frac{P[K^{-1}[m, n, 1]^T; 1]}{\|P[K^{-1}[m, n, 1]^T; 1]\|}, \quad X^P = t \tag{1}$$

Here, $X^K$ represents the normalized ray directions derived from camera intrinsics transformed by the pose $P$, and $X^P$ encodes the translational component $t$ of the pose. The homogeneous coordinate $[m, n, 1]^T$ is transformed by the inverse intrinsic matrix $K^{-1}$, projected via $P$, and normalized to ensure the resulting ray directions have unit length.

In cases where only camera intrinsics are available, encoding is performed in the local camera coordinate system:

$$X^K = \frac{K^{-1}[m, n, 1]^T}{\|K^{-1}[m, n, 1]^T\|} \tag{2}$$

This produces $X^K \in \mathbb{R}^{3 \times H \times W}$, representing ray directions in the camera's local frame.

Depth maps $D \in \mathbb{R}^{H \times W}$ are normalized in the range of $[0, 1]$ and paired with the corresponding binary masks $M \in \mathbb{R}^{H \times W}$ to form a composite representation, concatenated channel-wise:

$$X^D = [D; M] \tag{3}$$

The resulting $X^D \in \mathbb{R}^{2 \times H \times W}$ encapsulates depth values and their validity masks, enabling robust handling of sparse or incomplete depth data prevalent in real-world sensor outputs.

To prepare modalities for fusion, each is processed through a dedicated convolutional layer to extract initial feature maps, aligning their representations within a shared feature space:

$$
\begin{aligned}
F^D &= \text{Conv}_D(X^D), \\
F^K &= \text{Conv}_K(X^K), \\
F^P &= \text{Conv}_P(X^P),
\end{aligned}
\tag{4}
$$

where $\text{Conv}_D$, $\text{Conv}_K$, and $\text{Conv}_P$ are modality-specific convolutional layers tailored to the input dimensions and characteristics of $X^D$, $X^K$, and $X^P$, respectively. These layers produce feature maps $F^D, F^K, F^P \in \mathbb{R}^{C \times H \times W}$, where $C$ denotes the number of output channels.

**Modality Fusion.** We perform fusion five times within the CUT3R decoder, with the first fusion occurring before the initial decoder layer and subsequent fusions following each of the first four decoder layers. The features $F^D$, $F^K$, and $F^P$ are processed by dedicated ViT encoders, each comprising four layers, to extract intermediate representations tailored to the geometric and semantic properties of each modality. These encoders are not shared between modalities to preserve the unique characteristics of each modality.

To integrate the features from additional modalities $F^D$, $F^K$, and $F^P$ with the RGB image features $F^I$, we compute a guidance feature $G$ by summing the modality-specific features:

$$G = F^D + F^K + F^P \tag{5}$$

The guidance feature $G$ is combined with the RGB image features $F^I$ using a ZeroConv layer Zhang et al. (2023), a $1 \times 1$ convolution layer initialized with zero weights:

$$F^{\text{fused}} = F^I + \text{ZeroConv}(G) \tag{6}$$

The zero-initialized weights ensure that, at the very beginning of the training process, the additional modalities do not disrupt the pre-trained behavior of the CUT3R decoder. During fine-tuning, the model gradually learns to incorporate the guidance features. This allows the model to improve performance without destabilizing existing weights. This approach ensures a stable and effective integration of multimodal inputs, enhancing the robustness and adaptability of the model in complex vision tasks.

### 3.3 TRAINING OBJECTIVE

Our model predicts pointmaps $\hat{X} \in \mathbb{R}^{3 \times H \times W}$, the corresponding confidences $\hat{C} \in \mathbb{R}^{H \times W}$, and camera poses $\hat{P}$. Following the CUT3R framework (Wang et al., 2025b), the training objective comprises two primary components: a pointmap prediction loss $L_{\text{point}}$ and a camera pose prediction loss $L_{\text{pose}}$. The pose loss separately evaluates orientation error (via quaternion difference) and translation error, ensuring precise alignment of the predicted and the ground truth poses.

The loss functions are defined as follows:

$$
\begin{aligned}
L_{\text{point}} &= \sum_I \left( \hat{C} \| \hat{X} - X \| - \alpha \log \hat{C} \right), \\
L_{\text{pose}} &= \sum_I \left( \| \hat{q} - q \| + \| \hat{t} - t \| \right),
\end{aligned}
\tag{7}
$$

where $X$ and $\hat{X}$ represent the ground truth and predicted pointmaps, respectively, $\hat{C}$ denotes the predicted confidences, and $\alpha$ is a hyperparameter controlling the weight of the confidence regularization term. For pose loss, $\hat{q}$ and $q$ are the predicted and the ground truth quaternions, respectively, and $\hat{t}$ and $t$ are the predicted and ground truth translations. This composite loss ensures accurate point map reconstruction and robust pose estimation, which are critical for tasks such as 3D scene understanding and camera localization.

### 3.4 TRAINING STRATEGY

Our training strategy builds upon the CUT3R framework, utilizing short sequences of four images to ensure both computational efficiency and scalability. Unlike approaches that train separate models for each input modality (e.g., depth priors or camera parameters), we adopt a unified training paradigm. A single model is trained to handle arbitrary combinations of modalities, enhancing its versatility in real-world scenarios. During training, the model is exposed to random subsets of available modalities, simulating diverse input conditions. While modality-specific models may exhibit faster initial convergence, our unified model achieves comparable performance with sufficient training iterations, offering superior flexibility and practical deployment benefits.

**Datasets.** The model is trained on a diverse set of indoor and outdoor datasets, and the corresponding tables are provided in Supplementary Material. These include the Waymo Open Dataset (Sun et al., 2020), Co3Dv2 (Reizenstein et al., 2021), ScanNet (Dai et al., 2017), ARKitScenes (Baruch et al., 2021), DL3DV (Ling et al., 2024), WildRGBD (Xia et al., 2024), MegaDepth (Li & Snavely, 2018), ScanNet++ (Yeshwanth et al., 2023), MapFree (Arnold et al., 2022), TartanAir (Wang et al., 2020), BlendedMVS (Yao et al., 2020) and HyperSim (Roberts et al., 2021). Dataset preparation follows the same protocol as CUT3R to ensure consistency. To balance representation across datasets, we sample equal subsets of 10,000 examples from each, creating a comprehensive and diverse training corpus that supports robust generalization across indoor and outdoor scenes.

**Implementation details.** We employ a ViT-Large model (Dosovitskiy et al., 2021) as the image encoder and ViT-Base for all decoders. Each modality encoder consists of four transformer blocks with 12 attention heads and an embedding dimension of 768, striking a balance between feature richness and computational efficiency. Architectural parameters, such as a $16 \times 16$ patch size and embedding dimension, are aligned with those of the CUT3R RGB encoder to ensure compatibility. Linear layers serve as output heads to predict pointmaps, confidences, and camera poses. The model parameters are initialized using pre-trained CUT3R weights for images of size 512. We train the model using the Adam-W optimizer with a learning rate of $10^{-5}$, which incorporates a linear warmup and cosine weight decay schedule. The model is trained on four NVIDIA A100 GPUs for ten days, ensuring robust convergence and scalability.

## 4 EXPERIMENTS

We evaluate G-CUT3R across three tasks: scene-level 3D reconstruction, video depth estimation, and relative pose estimation, supported by thorough ablation studies and efficiency analyses. For all evaluations, we assess the impact of guidance using all possible combinations of auxiliary inputs, including camera intrinsics $K$, camera pose $R \,|\, t$, and depthmaps $D$. Also, we present results for two image resolutions, 224 and 512, demonstrating substantial performance improvements at both resolutions. Bold indicates the best performance, and underlined indicates the second best in the tables. Additionally, G-CUT3R demonstrates strong robustness to noisy priors, with detailed results for varying noise levels reported in Supplementary Material.

### 4.1 BENCHMARK SUITES

We evaluate our method on diverse datasets: *7-Scenes* (Shotton et al., 2013) and *NRGBD* (Azinović et al., 2022) for indoor static scenes with 3–5 views per scene following the low-overlap protocol of Wang et al. (2025b), *Bonn* (Palazzolo et al., 2019) for indoor dynamic scenes with handheld RGB-D sequences and highly dynamic objects involving human activities, and *Waymo* (Sun et al., 2020) for outdoor dynamic scenarios with moving agents and LiDAR depth data. Training, validation, and test splits adhere to the official splits provided by each dataset.

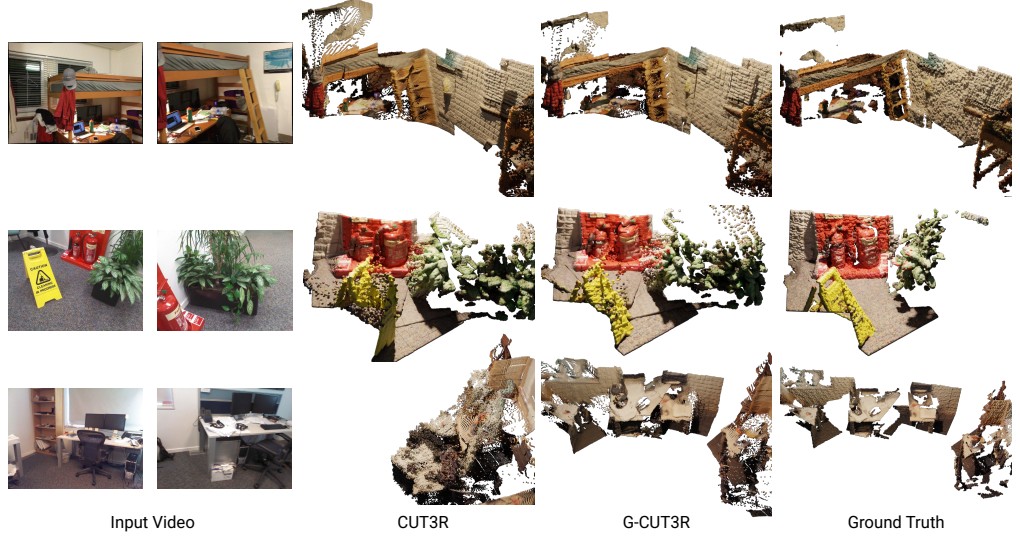

Input Video CUT3R G-CUT3R Ground Truth

Figure 3: Qualitative results on video sequences from the 7-scenes and ScanNet datasets. We compare our G-CUT3R method with CUT3R, demonstrating superior visual quality and reconstruction accuracy.

Table 1: 3D reconstruction comparison on 7-scenes and NRGBD datasets.

| Method | Resolution | FPS | K | R,t | D | 7-scenes Acc ↓ Mean | Med. | Comp ↓ Mean | Med. | NC ↑ Mean | Med. | NRGBD Acc ↓ Mean | Med. | Comp ↓ Mean | Med. | NC ↑ Mean | Med. |
|---|---|---|---|---|---|---|---|---|---|---|---|---|---|---|---|---|---|
| Spann3R | 224 | 16.3 | - | - | - | 0.298 | 0.226 | 0.205 | 0.112 | 0.650 | 0.730 | 0.416 | 0.323 | 0.417 | 0.285 | 0.684 | 0.789 |
| CUT3R | 224 | 33 | - | - | - | 0.298 | 0.203 | 0.254 | 0.110 | 0.649 | 0.728 | 0.422 | 0.213 | 0.252 | 0.163 | 0.713 | 0.835 |
| CUT3R | 512 | 20 | - | - | - | 0.126 | 0.047 | 0.154 | 0.031 | 0.727 | 0.834 | 0.099 | 0.031 | 0.076 | 0.026 | 0.837 | 0.971 |
| DUSt3R-GA | 512 | 0.9 | - | - | - | 0.146 | 0.077 | 0.181 | 0.067 | 0.736 | 0.839 | 0.144 | 0.019 | 0.154 | 0.018 | 0.870 | 0.982 |
| MASt3R-GA | 512 | 0.37 | - | - | - | 0.185 | 0.081 | 0.180 | 0.069 | 0.701 | 0.792 | 0.085 | 0.033 | 0.063 | 0.028 | 0.794 | 0.928 |
| Pow3R | 512 | 0.3 | - | - | - | 0.198 | 0.126 | 0.198 | 0.115 | 0.677 | 0.748 | 0.335 | 0.226 | 0.356 | 0.213 | 0.729 | 0.819 |
| Pow3R | 512 | 0.3 | + | - | - | 0.181 | 0.110 | 0.174 | 0.092 | 0.700 | 0.778 | 0.336 | 0.224 | 0.355 | 0.211 | 0.731 | 0.821 |
| Pow3R | 512 | 0.3 | - | + | - | 0.131 | 0.098 | 0.150 | 0.108 | 0.710 | 0.799 | 0.338 | 0.255 | 0.326 | 0.252 | 0.732 | 0.801 |
| Pow3R | 512 | 0.3 | - | - | + | 0.231 | 0.166 | 0.240 | 0.155 | 0.662 | 0.732 | 0.383 | 0.255 | 0.406 | 0.262 | 0.686 | 0.743 |
| Pow3R | 512 | 0.3 | + | + | - | 0.157 | 0.119 | 0.178 | 0.140 | 0.709 | 0.789 | 0.339 | 0.258 | 0.319 | 0.244 | 0.735 | 0.802 |
| Pow3R | 512 | 0.3 | + | - | + | 0.248 | 0.178 | 0.271 | 0.185 | 0.666 | 0.737 | 0.353 | 0.216 | 0.368 | 0.223 | 0.733 | 0.793 |
| Pow3R | 512 | 0.3 | - | + | + | 0.141 | 0.107 | 0.189 | 0.147 | 0.706 | 0.786 | 0.356 | 0.252 | 0.339 | 0.259 | 0.698 | 0.747 |
| Pow3R | 512 | 0.3 | + | + | + | 0.112 | 0.088 | 0.149 | 0.120 | 0.739 | 0.823 | 0.334 | 0.236 | 0.313 | 0.243 | 0.737 | 0.779 |
| G-CUT3R | 224 | 24 | - | - | - | 0.326 | 0.262 | 0.207 | 0.171 | 0.663 | 0.742 | 0.246 | 0.145 | 0.195 | 0.097 | 0.708 | 0.829 |
| G-CUT3R | 224 | 22.3 | + | - | - | 0.347 | 0.277 | 0.220 | 0.191 | 0.662 | 0.738 | 0.191 | 0.116 | 0.173 | 0.081 | 0.714 | 0.840 |
| G-CUT3R | 224 | 23.5 | - | + | - | 0.236 | 0.184 | 0.167 | 0.104 | 0.667 | 0.751 | 0.178 | 0.084 | 0.146 | 0.071 | 0.741 | 0.878 |
| G-CUT3R | 224 | 23.3 | - | - | + | 0.313 | 0.246 | 0.195 | 0.166 | 0.690 | 0.778 | 0.249 | 0.089 | 0.148 | 0.055 | 0.746 | 0.889 |
| G-CUT3R | 224 | 22.9 | + | + | - | 0.151 | 0.101 | 0.115 | 0.055 | 0.675 | 0.761 | 0.172 | 0.074 | 0.149 | 0.062 | 0.745 | 0.886 |
| G-CUT3R | 224 | 21.7 | + | - | + | 0.317 | 0.244 | 0.199 | 0.168 | 0.693 | 0.781 | 0.228 | 0.099 | 0.142 | 0.053 | 0.731 | 0.857 |
| G-CUT3R | 224 | 23.0 | - | + | + | 0.238 | 0.168 | 0.143 | 0.100 | 0.694 | 0.785 | 0.157 | 0.054 | 0.130 | 0.037 | 0.767 | 0.913 |
| G-CUT3R | 224 | 22.1 | + | + | + | 0.144 | 0.085 | 0.091 | 0.050 | 0.695 | 0.787 | 0.167 | 0.052 | 0.130 | 0.037 | 0.769 | 0.913 |
| G-CUT3R | 512 | 18 | - | - | - | 0.098 | 0.050 | 0.106 | 0.046 | 0.726 | 0.832 | 0.089 | 0.031 | 0.073 | 0.025 | 0.827 | 0.962 |
| G-CUT3R | 512 | 16.3 | + | - | - | 0.105 | 0.049 | 0.143 | 0.039 | 0.722 | 0.825 | 0.091 | 0.033 | 0.074 | 0.025 | 0.827 | 0.962 |
| G-CUT3R | 512 | 17.5 | - | + | - | 0.061 | 0.038 | 0.075 | 0.034 | 0.736 | 0.845 | 0.085 | 0.031 | 0.069 | 0.026 | 0.827 | 0.965 |
| G-CUT3R | 512 | 18 | - | - | + | 0.085 | 0.047 | 0.080 | 0.037 | 0.733 | 0.842 | 0.104 | 0.030 | 0.065 | 0.024 | 0.825 | 0.963 |
| G-CUT3R | 512 | 13.6 | + | + | - | 0.052 | 0.032 | 0.064 | 0.028 | 0.741 | 0.853 | 0.087 | 0.031 | 0.070 | 0.025 | 0.828 | 0.965 |
| G-CUT3R | 512 | 13.8 | + | - | + | 0.097 | 0.048 | 0.098 | 0.035 | 0.733 | 0.841 | 0.106 | 0.031 | 0.066 | 0.023 | 0.825 | 0.963 |
| G-CUT3R | 512 | 15.2 | - | + | + | 0.061 | 0.038 | 0.068 | 0.033 | 0.738 | 0.846 | 0.099 | 0.031 | 0.060 | 0.025 | 0.827 | 0.966 |
| G-CUT3R | 512 | 14.7 | + | + | + | **0.048** | **0.029** | **0.056** | **0.025** | 0.746 | **0.860** | 0.101 | 0.031 | 0.061 | 0.025 | 0.828 | 0.966 |

## 4.2 BASELINES

We evaluate our G-CUT3R method against leading dense 3D reconstruction approaches, including two-view and global optimization methods like DUSt3R (Wang et al., 2024b), which pioneered pointmap regression, MASt3R (Leroy et al., 2024), its successor with an added matching term, and Pow3R (Jang et al., 2025), our main competitor built on DUSt3R. We also compare against sequential reconstruction methods, including Spann3R (Wang & Agapito, 2024), featuring spatial memory, and CUT3R (Wang et al., 2025b), our primary backbone. Metrics are reported at 512 resolution for all methods except Spann3R, with CUT3R and G-CUT3R evaluated at both 224 and 512 resolutions.

## 4.3 3D RECONSTRUCTION

We evaluate scene-level 3D reconstruction on the *7-scenes* and *NRGBD* datasets using standard metrics: Accuracy (Acc), Completeness (Comp), and Normal Consistency (NC), as reported in

Table 2: Video depth evaluation on Bonn and ScanNet.

| Method | Resolution | FPS | $K$ | $R,t$ | Bonn | | ScanNet | |
|---|---|---|---|---|---|---|---|---|
| | | | | | Abs. rel ↓ | $\delta < 1.25$ ↑ | Abs. rel ↓ | $\delta < 1.25$ ↑ |
| Spann3R | 224 | 16.3 | - | - | 0.144 | 81.3 | 0.051 | 96.7 |
| CUT3R | 224 | 33 | - | - | 0.109 | 88.8 | 0.039 | 98.6 |
| CUT3R | 512 | 20 | - | - | 0.069 | 97.1 | _0.029_ | **99.3** |
| Pow3R | 512 | 0.3 | - | - | 0.148 | 67.1 | **0.028** | 99.0 |
| Pow3R | 512 | 0.3 | + | - | 0.132 | 75.2 | 0.038 | _99.2_ |
| Pow3R | 512 | 0.3 | - | + | 0.152 | 67.0 | **0.028** | 99.0 |
| Pow3R | 512 | 0.3 | + | + | 0.128 | 77.0 | 0.035 | _99.2_ |
| G-CUT3R | 224 | 24 | - | - | 0.126 | 89.9 | 0.04 | 98.5 |
| G-CUT3R | 224 | 22.3 | + | - | 0.125 | 85.8 | 0.04 | 98.5 |
| G-CUT3R | 224 | 23.5 | - | + | 0.105 | 89.1 | 0.04 | 98.6 |
| G-CUT3R | 224 | 22.9 | + | + | 0.104 | 88.4 | 0.04 | 98.7 |
| G-CUT3R | 512 | 18 | - | - | **0.062** | **97.5** | 0.031 | _99.2_ |
| G-CUT3R | 512 | 16.3 | + | - | _0.063_ | **97.5** | 0.031 | _99.2_ |
| G-CUT3R | 512 | 17.5 | - | + | _0.063_ | 97.4 | _0.029_ | _99.2_ |
| G-CUT3R | 512 | 13.6 | + | + | _0.063_ | 97.4 | 0.030 | _99.2_ |

Tab. 1. Following prior works (Leroy et al., 2024; Wang et al., 2025b), these metrics assess the quality of 3D scene reconstruction.

Consistent with CUT3R (Wang et al., 2025b), we assess performance under low-overlap conditions, where each scene comprises only 3–5 images, simulating challenging real-world scenarios with sparse viewpoints. The reported FPS was measured on NVIDIA A40 on $348 \times 512$ image resolution.

A discrepancy exists between the original CUT3R checkpoint and our unguided G-CUT3R variant due to differences in training data. Specifically, G-CUT3R is initialized from CUT3R checkpoints but fine-tuned on a smaller subset of the original training datasets, constrained by data availability. Consequently, a direct comparison with the original CUT3R would be biased, as its superior performance may result from exposure to a larger training corpus. To isolate the effect of guidance, we train a G-CUT3R variant without guidance on the same subset, ensuring a fair baseline for comparison.

As shown in Tab. 1, incorporating guidance consistently improves performance across both datasets and on both input resolutions. Camera poses contribute the most to enhancements in Accuracy and Completeness, while depth fusion significantly improves Normal Consistency. The fusion of multiple modalities outperforms single-modality configurations.

G-CUT3R offers the best accuracy–speed trade-off among all compared methods. It achieves the highest scores on 7-scenes and remains competitive on NRGBD. In contrast to slow optimization-based approaches such as DUSt3R-GA and MASt3R-GA, G-CUT3R runs at 13–18 FPS. Most notably, it substantially outperforms the closest competitor, Pow3R, in both reconstruction quality and speed.

## 4.4 VIDEO DEPTH ESTIMATION

We evaluate depth quality and consistency in video depth estimation on *Bonn* and *ScanNet* datasets for data sequences of length 10 using Absolute Relative Error (Abs. Rel) and the percentage of inlier points ($\delta < 1.25$), following established methods (Wang et al., 2025b; Zhang et al., 2024). For CUT3R and G-CUT3R, metrics are computed without scale alignment, because they estimate metric depth. Whereas for all other methods scale alignment is applied. Results are reported in Tab. 2. By integrating pose priors, our G-CUT3R method achieves improved Abs. Rel metric on the Bonn dataset across both image resolutions and beats all other methods. On the ScanNet dataset, G-CUT3R performs similarly to the best performing alternative approaches.

## 4.5 CAMERA POSE ESTIMATION

**Impact of priors.** Incorporating *pose* guidance significantly reduces ATE by **61%** on Sintel (from 0.077 to 0.030), **23%** on TUM RGB-D (from 0.013 to 0.010), and **29%** on ScanNet (from 0.007 to 0.005) compared to the no-guidance variant. Depth or intrinsic priors alone provide marginal improvements, but when combined with pose guidance, they further decrease RRE by 8–12% across all datasets, enhancing local pose accuracy.

## 4.6 ABLATION STUDY

Table 3: 3D reconstruction comparison of our method, both with and without zero convolutions, alongside an adaptation incorporating prior information in CUT3R inspired by Pow3R. Evaluations are performed on Waymo and ScanNet++ datasets, using the $L_2$ ↓ metric to assess reconstruction quality for four consecutive views.

| Method | $K$ | $R, t$ | $D$ | Waymo | | | | ScanNet++ | | | |
|---|---|---|---|---|---|---|---|---|---|---|---|
| | | | | $L_2/1$ | $L_2/2$ | $L_2/3$ | $L_2/4$ | $L_2/1$ | $L_2/2$ | $L_2/3$ | $L_2/4$ |
| Pow3R$^\dagger$ | - | - | - | 1.194 | 1.216 | 1.312 | 1.458 | 0.050 | 0.071 | 0.077 | 0.087 |
| Pow3R$^\dagger$ | + | - | - | 1.196 | 1.192 | 1.262 | 1.342 | 0.051 | 0.079 | 0.085 | 0.092 |
| Pow3R$^\dagger$ | - | + | - | 1.235 | 1.350 | 1.411 | 1.611 | 0.051 | 0.074 | 0.076 | 0.086 |
| Pow3R$^\dagger$ | - | - | + | 1.190 | 1.201 | 1.244 | 1.326 | 0.049 | 0.073 | 0.080 | 0.085 |
| Pow3R$^\dagger$ | + | + | - | 1.232 | 1.301 | 1.385 | 1.553 | 0.050 | 0.072 | 0.085 | 0.091 |
| Pow3R$^\dagger$ | + | - | + | 1.197 | 1.189 | 1.225 | 1.350 | 0.050 | 0.082 | 0.089 | 0.092 |
| Pow3R$^\dagger$ | - | + | + | 1.233 | 1.344 | 1.442 | 1.554 | 0.050 | 0.089 | 0.093 | 0.097 |
| Pow3R$^\dagger$ | + | + | + | 1.237 | 1.291 | 1.404 | 1.543 | 0.050 | 0.074 | 0.081 | 0.084 |
| Ours (w/o ZeroConv) | - | - | - | 1.796 | 1.723 | 1.756 | 1.766 | 0.055 | 0.088 | 0.092 | 0.100 |
| Ours (w/o ZeroConv) | + | - | - | 1.796 | 1.736 | 1.761 | 1.772 | 0.056 | 0.086 | 0.092 | 0.100 |
| Ours (w/o ZeroConv) | - | + | - | 1.798 | 1.721 | 1.803 | 2.006 | 0.056 | 0.072 | 0.083 | 0.081 |
| Ours (w/o ZeroConv) | - | - | + | 1.723 | 1.667 | 1.776 | 1.814 | 0.053 | 0.087 | 0.091 | 0.099 |
| Ours (w/o ZeroConv) | + | + | - | 1.797 | 1.723 | 1.802 | 2.002 | 0.056 | 0.073 | 0.081 | 0.080 |
| Ours (w/o ZeroConv) | + | - | + | 1.722 | 1.672 | 1.800 | 1.816 | 0.053 | 0.087 | 0.091 | 0.097 |
| Ours (w/o ZeroConv) | - | + | + | 1.734 | 1.667 | 1.776 | 1.965 | 0.053 | 0.072 | 0.082 | 0.079 |
| Ours (w/o ZeroConv) | + | + | + | 1.730 | 1.665 | 1.773 | 1.959 | 0.053 | 0.072 | 0.079 | 0.078 |
| Ours (w/ ZeroConv) | - | - | - | 1.235 | 1.259 | 1.300 | 1.327 | 0.049 | 0.074 | 0.075 | 0.086 |
| Ours (w/ ZeroConv) | + | - | - | 1.236 | 1.259 | 1.297 | 1.322 | 0.049 | 0.074 | 0.075 | 0.086 |
| Ours (w/ ZeroConv) | - | + | - | 1.235 | 1.215 | 1.248 | 1.305 | 0.049 | 0.066 | 0.067 | 0.070 |
| Ours (w/ ZeroConv) | - | - | + | **1.042** | 1.095 | 1.145 | 1.181 | **0.042** | **0.060** | **0.061** | **0.063** |
| Ours (w/ ZeroConv) | + | + | - | 1.235 | 1.215 | 1.246 | 1.301 | 0.049 | 0.067 | 0.067 | 0.070 |
| Ours (w/ ZeroConv) | + | - | + | **1.042** | 1.097 | 1.142 | 1.176 | **0.042** | 0.068 | 0.069 | 0.080 |
| Ours (w/ ZeroConv) | - | + | + | **1.042** | **1.054** | 1.091 | 1.159 | **0.042** | **0.060** | **0.061** | **0.063** |
| Ours (w/ ZeroConv) | + | + | + | **1.042** | 1.055 | **1.089** | **1.155** | **0.042** | 0.061 | **0.061** | 0.064 |

The original Pow3R implementation (Jang et al., 2025) is designed specifically for the DUSt3R architecture. To evaluate the effectiveness of our design choices, we adapted Pow3R (denoted as Pow3R$^\dagger$) to incorporate prior information into the CUT3R framework and conducted a direct comparison with our G-CUT3R method.

To ensure a fair comparison, both our method—with and without zero convolutions—and the Pow3R adaptation (Jang et al., 2025) are trained on the same dataset subset, comprising Waymo (Sun et al., 2020) and ScanNet++ (Yeshwanth et al., 2023), for an equal number of epochs. We evaluate the $L_2$ distance between ground truth and reconstructed pointmaps. Unlike standard Accuracy and Completeness metrics, which measure distances to the nearest points, the $L_2$ metric computes distances between points corresponding to identical pixel coordinates, offering a complementary perspective on reconstruction quality. The results are presented in Tab. 3.

Our ablation study demonstrates two key findings. First, the use of zero convolution layers significantly enhances reconstruction performance across metrics. Second, our approach to integrating prior information outperforms the Pow3R adaptation, attributed to its modality-agnostic fusion strategy and effective utilization of diverse input modalities.

Additionally, we ablate the choice of separate vs. shared encoders per modality and report results in Supplementary Material.

## 5 CONCLUSION

In this work, we present G-CUT3R, a lightweight and modality-agnostic extension to the CUT3R framework that enhances 3D scene reconstruction by integrating geometric priors such as camera calibrations, poses, and depth data. Our method employs straightforward encoding and carefully designed fusion during decoding. When tested on various datasets, G-CUT3R shows clear improvements over the existing state-of-the-art methods, producing more accurate and detailed 3D reconstructions. Our experiments further confirm that our fusion approach and design choices lead to better performance compared to other methods, making G-CUT3R a versatile and robust solution for 3D vision tasks.

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

## A  APPENDIX

This supplementary material provides additional details to support the main findings of our G-CUT3R model. We include comprehensive information on our training dataset, data preprocessing and camera pose estimation evaluation, along with the source code provided for G-CUT3R.

## B  TRAINING DATASET

Our training dataset consists of 12 datasets, including indoor/outdoor, real/synthetic, and dynamic/static variability. The complete list, together with pose and depth sources is provided in the Tab. 4

Table 4: Datasets for fine-tuning.

| Dataset Name | Scene Type | Dynamic? | Depth Source | Pose Source |
|---|---|---|---|---|
| ScanNet | Indoor | Static | RGBD (Microsoft Kinect v1) | SLAM |
| ScanNet++ | Indoor | Static | RGBD (Faro scanner + iPhone LiDAR) | SLAM |
| ARKitScenes | Indoor | Static | RGBD (iPhone/iPad LiDAR) | SLAM |
| Waymo | Outdoor | Dynamic | 64-beam LiDAR | SLAM |
| MegaDepth | Outdoor | Static | Multi-View Stereo (MVS) | SLAM |
| DL3DV | Mixed | Static | Multi-View Stereo (MVS) | SLAM |
| Co3Dv2 | Object-centric | Static | Multi-View Stereo (MVS) | SLAM |
| WildRGBD | Object-centric | Static | RGBD (iPhone/iPad LiDAR) | SLAM |
| MapFree | Outdoor | Static | Multi-View Stereo (MVS) | SLAM |
| TartanAir | Mixed (Synthetic) | Dynamic | Rendering | GT |
| BlendedMVS | Mixed (Synthetic) | Static | Multi-View Stereo (MVS) | SLAM |
| HyperSim | Indoor (Synthetic) | Static | Rendering | GT |

## C TRAINING DATASET PREPROCESSING

Our training dataset is derived from a subset of the CUT3R dataset (Wang et al., 2025b), comprising both ordered and unordered image sequences. For ordered sequences, we sample frames at random intervals ranging from 1 to $k$, where $k$ varies by dataset. For unordered images, we select samples based on their visual overlap. For the DL3DV dataset (Ling et al., 2024), we utilize the depth maps provided by the CUT3R authors. The preprocessing pipeline follows the methodology outlined in CUT3R (Wang et al., 2025b), and we refer readers to the original paper for further details.

## D CAMERA POSE ESTIMATION

Following the evaluation protocol of Zhao et al. (2022); Wang et al. (2025b), we align each predicted trajectory to the ground truth using a seven-degree-of-freedom similarity transform and report the following metrics: Absolute Translation Error (ATE) – global drift of the entire trajectory; Relative Translation Error (RTE) – average translational drift between consecutive frames; Relative Rotation Error (RRE) – average rotational drift between consecutive frames. Experiments are conducted on the *Sintel* (Butler et al., 2012), *TUM dynamics* (Sturm et al., 2012), and *ScanNet* (Dai et al., 2017) datasets.

Table 5: Evaluation on camera pose estimation on Sintel, TUM dynamics and ScanNet datasets.

| Method | Resolution | FPS | K | D | Sintel ATE ↓ | Sintel RPE trans ↓ | Sintel RPE rot ↓ | TUM dynamics ATE ↓ | TUM dynamics RPE trans ↓ | TUM dynamics RPE rot ↓ | ScanNet ATE ↓ | ScanNet RPE trans ↓ | ScanNet RPE rot ↓ |
|---|---|---|---|---|---|---|---|---|---|---|---|---|---|
| Spann3R | 224 | 16.3 | - | - | 0.329 | **0.110** | 4.471 | 0.056 | 0.021 | 0.591 | 0.096 | 0.023 | 0.661 |
| CUT3R | 224 | 33 | - | - | 0.090 | 0.172 | 0.746 | 0.011 | 0.013 | 0.597 | 0.020 | 0.020 | 0.514 |
| CUT3R | 512 | 20 | - | - | 0.086 | 0.156 | 0.433 | **0.009** | 0.011 | 0.499 | 0.008 | 0.012 | 0.327 |
| Pow3R | 512 | 0.3 | - | - | 0.578 | 0.651 | 1.877 | 0.027 | 0.021 | 1.625 | 0.019 | 0.022 | 0.988 |
| Pow3R | 512 | 0.3 | + | - | 0.457 | 0.665 | 2.681 | 0.023 | 0.022 | 1.643 | 0.021 | 0.023 | 0.986 |
| Pow3R | 512 | 0.3 | - | + | 0.412 | 0.621 | 1.519 | 0.018 | 0.020 | 1.439 | 0.016 | 0.022 | 0.959 |
| Pow3R | 512 | 0.3 | + | + | 0.426 | 0.610 | 0.974 | 0.013 | 0.018 | 1.425 | 0.019 | 0.022 | 0.957 |
| G-CUT3R | 224 | 24 | - | - | 0.077 | 0.177 | 0.919 | 0.013 | 0.017 | 0.634 | **0.007** | **0.008** | 1.351 |
| G-CUT3R | 224 | 22.3 | + | - | 0.080 | 0.180 | 0.925 | 0.012 | 0.017 | 0.618 | **0.007** | **0.008** | 1.336 |
| G-CUT3R | 224 | 23.3 | - | + | 0.097 | 0.171 | 0.867 | 0.013 | 0.015 | 0.635 | **0.007** | **0.008** | 1.340 |
| G-CUT3R | 224 | 21.7 | + | + | 0.099 | 0.172 | 0.866 | 0.013 | 0.016 | 0.619 | **0.007** | **0.008** | 1.324 |
| G-CUT3R | 512 | 18 | - | - | 0.063 | 0.162 | 0.526 | 0.010 | 0.011 | 0.437 | 0.008 | 0.011 | 0.320 |
| G-CUT3R | 512 | 16.3 | + | - | 0.063 | 0.162 | 0.517 | 0.010 | 0.011 | 0.437 | 0.008 | 0.011 | **0.319** |
| G-CUT3R | 512 | 18 | - | + | 0.055 | 0.160 | 0.509 | **0.009** | **0.010** | 0.433 | 0.008 | 0.011 | 0.321 |
| G-CUT3R | 512 | 13.8 | + | + | **0.054** | 0.159 | **0.498** | **0.009** | **0.010** | **0.431** | 0.008 | 0.011 | **0.319** |

We evaluate relative pose estimation of our G-CUT3R method against Spann3R (Wang & Agapito, 2024), CUT3R (Wang et al., 2025b) and Pow3R (Jang et al., 2025) on the *Sintel* (Butler et al., 2012), *TUM dynamics* (Sturm et al., 2012), and *ScanNet* (Dai et al., 2017) datasets, employing Absolute Translation Error (ATE), Relative Translation Error (RPE trans), and Relative Rotation Error (RPE rot) as metrics, with trajectories aligned to ground truth following the protocol of Wang et al. (2025b). The results are presented in Tab. 5. Both - depth and intrinsics priors improve metrics on Sintel and TUM dynamics. On ScanNet the base model already demonstrates good results and performance improvement from depth and intrinsics priors is negligible.

Table 6: 3D reconstruction comparison on 7-scenes and NRGBD datasets.

| Noise, % | 7-scenes Acc ↓ Mean | Acc ↓ Med. | Comp ↓ Mean | Comp ↓ Med. | NC ↑ Mean | NC ↑ Med. | NRGBD Acc ↓ Mean | Acc ↓ Med. | Comp ↓ Mean | Comp ↓ Med. | NC ↑ Mean | NC ↑ Med. |
|---|---|---|---|---|---|---|---|---|---|---|---|---|
| **G-CUT3R (Pose-guided)** | | | | | | | | | | | | |
| 0 | 0.065 | 0.040 | 0.067 | 0.034 | 0.720 | 0.828 | 0.091 | 0.037 | 0.064 | 0.028 | 0.826 | 0.967 |
| 5 | 0.066 | 0.039 | 0.067 | 0.034 | 0.720 | 0.829 | 0.095 | 0.040 | 0.067 | 0.030 | 0.821 | 0.965 |
| 10 | 0.063 | 0.036 | 0.063 | 0.031 | 0.723 | 0.833 | 0.106 | 0.048 | 0.073 | 0.034 | 0.818 | 0.961 |
| 20 | 0.067 | 0.040 | 0.065 | 0.033 | 0.721 | 0.830 | 0.096 | 0.043 | 0.066 | 0.029 | 0.817 | 0.965 |
| 30 | 0.082 | 0.048 | 0.075 | 0.038 | 0.718 | 0.825 | 0.111 | 0.056 | 0.080 | 0.039 | 0.808 | 0.956 |
| 40 | 0.082 | 0.051 | 0.072 | 0.035 | 0.718 | 0.825 | 0.115 | 0.082 | 0.106 | 0.064 | 0.787 | 0.918 |
| 50 | 0.112 | 0.054 | 0.095 | 0.036 | 0.714 | 0.822 | 0.118 | 0.078 | 0.100 | 0.060 | 0.786 | 0.934 |
| **G-CUT3R (Depth-guided)** | | | | | | | | | | | | |
| 0 | 0.092 | 0.047 | 0.071 | 0.033 | 0.720 | 0.828 | 0.102 | 0.038 | 0.062 | 0.027 | 0.824 | 0.964 |
| 5 | 0.091 | 0.048 | 0.070 | 0.033 | 0.719 | 0.827 | 0.098 | 0.037 | 0.063 | 0.026 | 0.825 | 0.965 |
| 10 | 0.091 | 0.047 | 0.071 | 0.033 | 0.719 | 0.826 | 0.091 | 0.036 | 0.064 | 0.026 | 0.826 | 0.965 |
| 20 | 0.091 | 0.049 | 0.070 | 0.034 | 0.715 | 0.822 | 0.088 | 0.036 | 0.065 | 0.026 | 0.826 | 0.965 |
| 30 | 0.119 | 0.053 | 0.132 | 0.034 | 0.698 | 0.798 | 0.090 | 0.038 | 0.068 | 0.027 | 0.824 | 0.964 |
| 40 | 0.117 | 0.052 | 0.134 | 0.035 | 0.701 | 0.800 | 0.091 | 0.039 | 0.070 | 0.028 | 0.823 | 0.964 |
| 50 | 0.122 | 0.054 | 0.133 | 0.034 | 0.702 | 0.803 | 0.091 | 0.039 | 0.071 | 0.028 | 0.822 | 0.964 |

# E  NOISE ROBUSTNESS ANALYSIS

We have evaluated the robustness of our method by injecting Gaussian noise directly into the additional prior modalities. The noise was sampled from a normal distribution with mean = 0 and std ranging from 5% to 50% of the ground truth data.

The results in Tab. 6 show that G-CUT3R consistently outperforms the non-guided baseline even at a noise level of up to 20%, and it degrades more significantly at higher noise. This confirms the robustness of our guided fusion mechanism to realistic sensor noise.

# F  ADDITIONAL ABLATIONS

We have compared two types of encoders (training and testing on the combined ScanNet++ and Waymo):

- one version with a shared VIT encoder for all guidance modalities
- one version with separate modality-specific encoders

Results in Tab. 7 show that the shared encoder performs very well on ScanNet++ but is clearly outperformed by separate encoders on the more challenging Waymo dataset (outdoor scenes, large scale, dynamic objects). On easier / indoor scenes the shared encoder remains competitive. This supports our design choice of separate encoders for the full method while indicating that a shared encoder can be a reasonable lighter alternative in simpler settings.

Table 7: Ablation results on ScanNet++ and Waymo datasets for different combinations of camera intrinsics ($K$), pose ($R, t$) and depth ($D$).

| Encoder | $K$ | $R, t$ | $D$ | ScanNet++ Acc ↓ Mean | Acc ↓ Med. | Comp ↓ Mean | Comp ↓ Med. | NC ↑ Mean | NC ↑ Med. | Waymo Acc ↓ Mean | Acc ↓ Med. | Comp ↓ Mean | Comp ↓ Med. | NC ↑ Mean | NC ↑ Med. |
|---|---|---|---|---|---|---|---|---|---|---|---|---|---|---|---|
| Separate | - | - | - | 0.142 | 0.104 | 0.090 | 0.052 | 0.707 | 0.805 | 0.753 | 0.482 | 1.154 | 0.463 | 0.709 | 0.822 |
| Separate | + | - | - | 0.143 | 0.107 | 0.092 | 0.055 | 0.706 | 0.803 | 0.754 | 0.476 | 1.152 | 0.461 | 0.709 | 0.822 |
| Separate | - | + | - | 0.136 | 0.099 | 0.087 | 0.050 | 0.704 | 0.800 | 0.704 | 0.456 | 1.095 | 0.431 | 0.704 | 0.817 |
| Separate | - | - | + | 0.130 | 0.090 | 0.075 | 0.037 | 0.717 | 0.816 | 0.709 | 0.472 | 0.870 | 0.416 | 0.712 | 0.825 |
| Separate | + | + | - | 0.128 | 0.091 | 0.082 | 0.044 | 0.705 | 0.802 | 0.712 | 0.466 | 1.103 | 0.443 | 0.704 | 0.816 |
| Separate | + | - | + | 0.129 | 0.094 | 0.077 | 0.041 | 0.715 | 0.812 | 0.732 | 0.501 | 0.891 | 0.444 | 0.711 | 0.825 |
| Separate | - | + | + | 0.121 | 0.079 | 0.081 | 0.039 | 0.722 | 0.825 | 0.701 | 0.462 | 0.847 | 0.403 | 0.710 | 0.825 |
| Separate | + | + | + | 0.113 | 0.072 | 0.077 | 0.031 | 0.723 | 0.826 | 0.682 | 0.444 | 0.832 | 0.389 | 0.711 | 0.825 |
| Shared | - | - | - | 0.146 | 0.105 | 0.090 | 0.056 | 0.708 | 0.810 | 0.760 | 0.508 | 1.210 | 0.485 | 0.705 | 0.818 |
| Shared | + | - | - | 0.147 | 0.105 | 0.090 | 0.055 | 0.708 | 0.810 | 0.757 | 0.504 | 1.207 | 0.480 | 0.705 | 0.818 |
| Shared | - | + | - | 0.133 | 0.089 | 0.091 | 0.047 | 0.710 | 0.814 | 0.725 | 0.456 | 1.135 | 0.441 | 0.707 | 0.818 |
| Shared | - | - | + | 0.132 | 0.098 | 0.077 | 0.046 | 0.711 | 0.811 | 0.831 | 0.602 | 1.004 | 0.553 | 0.706 | 0.821 |
| Shared | + | + | - | 0.130 | 0.088 | 0.092 | 0.045 | 0.711 | 0.814 | 0.725 | 0.459 | 1.141 | 0.446 | 0.708 | 0.819 |
| Shared | + | - | + | 0.133 | 0.099 | 0.077 | 0.047 | 0.713 | 0.813 | 0.796 | 0.569 | 0.976 | 0.522 | 0.707 | 0.822 |
| Shared | - | + | + | 0.123 | 0.078 | 0.080 | 0.036 | 0.719 | 0.822 | 0.711 | 0.465 | 0.874 | 0.406 | 0.709 | 0.822 |
| Shared | + | + | + | 0.118 | 0.077 | 0.081 | 0.035 | 0.720 | 0.824 | 0.736 | 0.498 | 0.900 | 0.447 | 0.708 | 0.821 |

