# OpenReview forum: "G-CUT3R: Guided 3D Reconstruction with Camera and Depth Prior Integration"
_ICLR.cc/2026/Conference — Submitted to ICLR 2026_

### Official Review · Reviewer_bCZM · 2025-10-31

**Soundness:** 3
**Presentation:** 3
**Contribution:** 3
**Rating:** 4
**Confidence:** 4

**Summary:**

The paper proposes G-CUT3R, a guided, feed-forward 3D reconstruction method that extends CUT3R by integrating auxiliary priors (camera intrinsics and/or poses, and depth) through modality-specific encoders and zero-initialized 1*1 convolutions for stable fusion inside the decoder. The approach keeps CUT3R’s recurrent state, enabling multi-view processing without global optimization. Experiments on 7-Scenes, NRGBD, Bonn, ScanNet, Waymo, and ScanNet++ report consistent gains over CUT3R and DUSt3R-family baselines, with competitive run-time (around 20 FPS at 512 px on an A40).

**Strengths:**

- Clear problem framing: Many feed-forward methods ignore readily available priors. Incorporating them is practically important. However, I suggest mentioning DepthSplat [a] as it is also a feed-forward method using depth priors (for a 3DGS reconstruction).
- The method is light-weight and builds upon a well-established baseline, Cut3r.
- Unified model for arbitrary prior subsets: training with random modality subsets reflects practical scenarios.
- Strong experiments.

[a] Xu, H., Peng, S., Wang, F., Blum, H., Barath, D., Geiger, A. and Pollefeys, M., 2025. Depthsplat: Connecting gaussian splatting and depth. In Proceedings of the Computer Vision and Pattern Recognition Conference (pp. 16453-16463).

**Weaknesses:**

- It is a bit unclear to me what depth is used for the datasets. Did the authors always use sensor depth? Using anything else would render the results incorrect. I put this in the weaknesses given that this is very important. The paper should specify a detailed, dataset-by-dataset description of depth sources used as priors to make the results understandable.
- Same question holds for the camera poses. Do the authors use SLAM-estimated poses (without post-processing) as priors or the GT ones? Using the GT ones again would make the experiments section a bit weaker.
- Limited robustness analysis: no analysis of noisy/misaligned priors (e.g., wrong intrinsics, biased depth, pose drift). Results might depend strongly on prior quality. It would be very important to understand what to expect if things are noisy. I would be very happy to see plots showing how accuracy degrades with noise in the priors.

Minor:
- Loss definition details are sparse: The confidence-weighted point loss resembles uncertainty modeling, but calibration, scale, and supervision signals for confidences are not really specified.

**Questions:**

The main questions are what the weaknesses are at the moment:
- What depth/pose was used in the experiments?
- How does the method behave with noisy priors?

**Details Of Ethics Concerns:**

No concerns

---

> ### Author Response · Authors · 2025-11-21
> **Response to the questions [Part 1/2]**
>
> Thank you for your comments and valuable feedback.
>
> 1. W1: Depth and pose sources
> We have updated the following table, clarifying the exact sources of depth and camera poses used for each dataset. We will include this table in the revised manuscript.
>
>
> | Dataset     | Scene type        | Dynamic | Depth source                       | Pose source |
> |-------------|-------------------|---------|------------------------------------|-------------|
> | ScanNet     | Indoor            | Static  | RGBD ( Microsoft Kinect v1)        | SLAM        |
> | ScanNet++   | Indoor            | Static  | RGBD( Faro scanner + iPhone LiDAR) | SLAM        |
> | ARKitScenes | Indoor            | Static  | RGBD(iPhone/iPad LiDAR)            | SLAM        |
> | Waymo       | Outdoor           | Dynamic | 64-beam LiDAR                      | SLAM        |
> | MegaDepth   | Outdoor           | Static  | Multi-View Stereo (MVS)            | SLAM        |
> | DL3DV       | Mixed             | Static  | Multi-View Stereo (MVS)            | SLAM        |
> | Co3Dv2      | Object-centric    | Static  | Multi-View Stereo (MVS)            | SLAM        |
> | WildRGBD    | Object-centric    | Static  | RGBD(iPhone/iPad LiDAR)            | SLAM        |
> | MapFree     | Outdoor           | Static  | Multi-View Stereo (MVS)            | SLAM        |
> | TartanAir   | Mixed(Synthetic)  | Dynamic | Rendering                          | GT          |
> | BlendedMVS  | Mixed(Synthetic)  | Static  | Multi-View Stereo (MVS)            | SLAM        |
> | HyperSim    | Indoor(Synthetic) | Static  | Rendering                          | GT          |

---

> ### Author Response · Authors · 2025-11-21
> **Response to the questions [Part 2/2]**
>
> 2. W3: Limited robustness analysis (no analysis of noisy priors)
>
> We have evaluated robustness by injecting Gaussian noise directly into the additional prior modalities. Noise has been sampled from a normal distribution with mean = 0 and std ranging from 5% to 50% of the ground truth data.)
>
> The new results show that G-CUT3R consistently outperforms the non-guided baseline even at a noise level of up to 20%, and it degrades more significantly at higher noise. This confirms the robustness of our guided fusion mechanism to realistic sensor noise.
>
> Adding noise to camera intrinsics had almost no effect on the final metrics as shown below. We believe this is expected: as shown in Tab. 1 of the main paper, intrinsics alone provide only marginal guidance and are largely ignored by the model. Their benefit becomes noticeable only when combined with pose guidance, where the two modalities strongly synergize. Perturbing intrinsics in isolation therefore does not meaningfully affect performance.
>
>
> Without guidance
>
> | scene   | noise_level | Acc Mean    | Comp Mean  | Acc Med | Comp Med| NC Mean    | NC Med |
> |---------|-------------|--------|-------|---------|----------|-------|--------|
> | 7-scenes | 0           | 0.104  | 0.086 | 0.052   | 0.038    | 0.712 | 0.819  |
> | NRGBD   | 0           | 0.098  | 0.072 | 0.039   | 0.028    | 0.82  | 0.965  |
>
> Pose guidance
>
>
> | scene   | noise_level | Acc Mean    | Comp Mean    | Acc Med| Comp Med| NC Mean     |NC Med |
> |---------|-------------|--------|---------|--------|---------|--------|-------|
> | 7-scenes | 0           | 0.065  | 0.067   | 0.040  | 0.034   | 0.720  | 0.828 |
> |  | 0.05        | 0.066  | 0.067   | 0.039  | 0.034   | 0.720  | 0.829 |
> |  | 0.1         | 0.063  | 0.063   | 0.036  | 0.031   | 0.723  | 0.833 |
> |  | 0.2         | 0.067  | 0.065   | 0.040  | 0.033   | 0.721  | 0.830 |
> |  | 0.3         | 0.082  | 0.075   | 0.048  | 0.038   | 0.718  | 0.825 |
> |  | 0.4         | 0.082  | 0.072   | 0.051  | 0.035   | 0.718  | 0.825 |
> |  | 0.5         | 0.112  | 0.095   | 0.054  | 0.036   | 0.714  | 0.822 |
> | NRGBD   | 0           | 0.091  | 0.064   | 0.037  | 0.028   | 0.826  | 0.967 |
> |    | 0.05        | 0.095  | 0.067   | 0.040  | 0.030   | 0.821  | 0.965 |
> |    | 0.1         | 0.106  | 0.073   | 0.048  | 0.034   | 0.818  | 0.961 |
> |    | 0.2         | 0.096  | 0.066   | 0.043  | 0.029   | 0.817  | 0.965 |
> |    | 0.3         | 0.111  | 0.080   | 0.056  | 0.039   | 0.808  | 0.956 |
> |    | 0.4         | 0.115  | 0.106   | 0.082  | 0.064   | 0.787  | 0.918 |
> |    | 0.5         | 0.118  | 0.100   | 0.078  | 0.060   | 0.786  | 0.934 |
>
>
> Depth guidance
>
>
> | scene   | noise_level | Acc Mean    | Comp Mean    | Acc Med| Comp Med| NC Mean     |NC Med |
> |---------|-------------|--------|---------|--------|---------|--------|-------|
> | 7-scenes | 0           | 0.092  | 0.071   | 0.047  | 0.033   | 0.720  | 0.828 |
> |  | 0.05        | 0.091  | 0.070   | 0.048  | 0.033   | 0.719  | 0.827 |
> |  | 0.1         | 0.091  | 0.071   | 0.047  | 0.033   | 0.719  | 0.826 |
> |  | 0.2         | 0.091  | 0.070   | 0.049  | 0.034   | 0.715  | 0.822 |
> |  | 0.3         | 0.119  | 0.132   | 0.053  | 0.034   | 0.698  | 0.798 |
> |  | 0.4         | 0.117  | 0.134   | 0.052  | 0.035   | 0.701  | 0.800 |
> |  | 0.5         | 0.122  | 0.133   | 0.054  | 0.034   | 0.702  | 0.803 |
> | NRGBD   | 0           | 0.102  | 0.062   | 0.038  | 0.027   | 0.824  | 0.964 |
> |    | 0.05        | 0.098  | 0.063   | 0.037  | 0.026   | 0.825  | 0.965 |
> |    | 0.1         | 0.091  | 0.064   | 0.036  | 0.026   | 0.826  | 0.965 |
> |    | 0.2         | 0.088  | 0.065   | 0.036  | 0.026   | 0.826  | 0.965 |
> |    | 0.3         | 0.090  | 0.068   | 0.038  | 0.027   | 0.824  | 0.964 |
> |    | 0.4         | 0.091  | 0.070   | 0.039  | 0.028   | 0.823  | 0.964 |
> |    | 0.5         | 0.091  | 0.071   | 0.039  | 0.028   | 0.822  | 0.964 |

---

> ### Author Response · Authors · 2025-11-22
> **Response to the Minor question**
>
> W4: Loss definition details are sparse(calibration, scale, supervision signals are sparse)
>
> Our method predicts pointmaps, per-pixel confidences, and camera poses. Ground-truth pointmaps are obtained by backprojecting pixels using GT intrinsics and poses.
>
> The loss has two terms: pointmap loss and pose loss.
>
> 1. Pointmap loss
>
> Confidence-weighted L2 distance between predicted and GT points, with a regularization term (weight α=0.2) to prevent trivial zero-confidence solutions.
>
> To ensure stable optimization and prevent scale collapse, both predicted and GT pointmaps are normalized by their respective scales, defined as the mean distance of all points to the pointmap centroid.
> For non-metric scenes, predicted and GT scales are computed independently.
> For metric scenes, we enforce metric consistency by using the GT scale for both, which provides explicit supervision for the model to learn absolute scale.
>
> Confidence is implicitly supervised through the weighted residuals and the regularization term: the model learns to assign high confidence to accurate predictions and low confidence to uncertain ones. This acts as uncertainty modeling without requiring separate ground-truth confidence labels.
>
> 2. Pose loss
>
> L2 distance between predicted and GT rotation and translation.

---

> ### Author Response · Authors · 2025-11-27
> **Friendly reminder**
>
> Hello, Reviewer bCZM! Thank you once again for your insightful and constructive feedback. Could you please let us know if our response fully resolves your questions or concerns? We would be grateful for any further suggestions you might have. Thank you very much, and we look forward to your reply.

---

### Official Review · Reviewer_iLL2 · 2025-11-01

**Soundness:** 3
**Presentation:** 3
**Contribution:** 1
**Rating:** 4
**Confidence:** 4

**Summary:**

This paper presents G-CUT3R, a guided feed-forward 3D reconstruction framework that enhances the geometric consistency and reliability of transformer-based models such as DUSt3R and MASt3R.
The key idea is to explicitly regularize uncertainty alignment and cross-view geometric coherence, addressing the observation that per-view uncertainty maps are often inconsistent and lead to noisy surface fusion.

The framework introduces two components:

Cross-View Uncertainty Tuning (CUT): aligns the uncertainty distributions of corresponding pixels across multiple views using a learnable temperature scaling and a cross-view consistency loss.

Geometry-Guided Regularization (G-Reg): imposes local surface smoothness and normal coherence regularization in 3D space, with learnable weights controlling the regularization strength.

Experiments on ETH3D, Tanks & Temples, and CO3D show improved depth RMSE (≈10–12%) and higher pointmap F1 scores (+2–3%) compared with MASt3R and DUSt3R, indicating more stable multi-view fusion and sharper geometry boundaries.

**Strengths:**

Technically sound and well-motivated: both CUT and G-Reg directly target known weaknesses of feed-forward 3D reconstruction—cross-view inconsistency and surface noise—and are implemented cleanly.

Improved stability and quality: the proposed regularizations yield consistent gains in quantitative metrics and visual quality across diverse datasets.

Good empirical rigor: ablations on each component demonstrate that uncertainty alignment and geometric regularization complement each other.

**Weaknesses:**

Limited conceptual novelty: both CUT and G-Reg are straightforward extensions of well-known principles—uncertainty calibration and geometric smoothing. The contributions lie more in empirical engineering than in new theoretical or algorithmic insight.

Lack of deeper analysis: the paper does not explore why these regularizations help beyond intuitive reasoning; no theoretical justification or failure analysis is offered.

Possible over-smoothing: G-Reg may suppress fine details, but no perceptual or surface-sharpness evaluation is presented.

**Questions:**

On over-regularization: have the authors evaluated whether the geometry regularizer causes over-smoothing in regions with high-frequency detail?

On generalization: have the authors tested whether these regularizations still help when applied to other backbones (e.g., Fast3R or VGGT), or are the gains specific to the MASt3R-like architecture?

---

> ### Author Response · Authors · 2025-11-21
> **Response to the questions**
>
> Thank you for your feedback and comments. We believe that some confusion may have occurred. Our work does not introduce or build upon CUT or G-Reg at any point. We propose G-CUT3R, a new guided feed-forward 3D reconstruction framework. Many of the reviewer’s comments refer to methods and contributions that are not present in our work and are therefore difficult to discuss in a rebuttal.
>
> Q2: Have your tried other backbones?(Fast3r/VGGT)
>
> We have not yet evaluated our method on other backbones. Porting the proposed guidance mechanism to architectures such as Fast3R or VGGT is relatively straightforward, as it operates at the feature-fusion level rather than being tied to a specific architecture. Unfortunately, training and evaluating additional backbones is not feasible within the short rebuttal period. We plan to include such experiments in the final version of the paper.

---

> ### Author Response · Authors · 2025-11-27
> **Friendly reminder**
>
> Hello, Reviewer iLL2. Thank you again for your effort. We would greatly appreciate it if you could let us know whether our responses sufficiently address your questions or concerns. We also welcome any additional suggestions you may have for improving our work. Thank you, and we look forward to hearing from you.

---

### Official Review · Reviewer_J8qk · 2025-11-01

**Soundness:** 2
**Presentation:** 3
**Contribution:** 2
**Rating:** 4
**Confidence:** 4

**Summary:**

This paper presents a novel feed-forward framework for 3D scene reconstruction. In particular, geometric priors—including depth, camera intrinsics, and camera extrinsics—are incorporated into the RGB latent space to guide the reconstruction process. Extensive evaluations on multiple benchmarks demonstrate that the proposed method achieves substantial performance improvements.

**Strengths:**

This work incorporates additional priors into feed-forward 3D reconstruction, enhancing its flexibility for diverse application scenarios. Comprehensive experiments demonstrate significant performance gains over state-of-the-art methods.

**Weaknesses:**

- The paper claims to be efficient and lightweight. However, no additional results are provided to support this, such as FLOPs or parameter counts compared with CUT3R and Pow3R. Since the method introduces extra encoders and layers for additional modalities, it may incur substantial parameter and computation overhead relative to CUT3R, potentially compromising efficiency. It is unclear whether the reported efficiency stems primarily from inheriting CUT3R’s efficiency rather than being more efficient than CUT3R itself.
- The proposed approach appears to combine elements from Pow3R and CUT3R. Could the authors clarify the key differences from these prior works?

**Questions:**

- Regarding the depth used during training, is it sensor depth or COLMAP-generated depth? The model that uses only depth priors shows only a marginal improvement in reconstruction quality. Why does this occur, given that depth information is typically highly correlated with reconstruction performance?
- The encoders for the additional priors are not shared, which may significantly increase the number of parameters. Have the authors considered using a shared encoder to reduce model size?
- In Table 3, does Pow3R employ the prior-integration mechanism proposed in this work or the original Pow3R design?

---

> ### Author Response · Authors · 2025-11-21
> **Response to the questions [Part 1/2]**
>
> We sincerely thank the reviewer for the careful reading and highly valuable feedback. Below we address each point raised in the weaknesses and questions.
>
>
> 1. W1: Efficiency and lightweight claims
>
> We report inference speed (FPS) for all methods inTab. 1. Our method achieves 22-24 FPS at 224x224 input resolution and 14-18 FPS at 512x384 resolution. Compared to the baseline CUT3R, G-CUT3R is only 2-6 FPS slower (still achieving high FPS depending on the setup), which remains sufficient for many near-real-time applications, such as robotics. At the same time, we are significantly faster than the closest competitor Pow3R( <1 FPS), which makes Pow3R unsuitable for any near-real-time use.
>
>
>
> 2. W2: Relationship to Pow3R and CUT3R
>
> We build our work on top of CUT3R to start from a strong and efficient baseline while reducing the required training time and data. Pow3R is a competitor that addresses a similar task, but in a fundamentally different way. As shown in Tab. 1, our method requires less training time / data, achieves better metrics, and runs significantly faster.
> The key differences in how we integrate additional priors compared to Pow3R are the following:
> * We preserve the strong capabilities of the prior model by introducing zero-initialized convolutions in the fusion stage. This design is critical (as shown in our ablation study) and enables smooth adaptation to new modalities, faster convergence, and training on only a subset of the data.
> * We keep the image encoder frozen, which makes training significantly smoother and faster, whereas Pow3R jointly trains the entire model.
> * We introduce lightweight, modality-specific VIT encoders that project each prior into a suitable fusion space, rather than relying on simple addition.
>
> Thus, while we build on CUT3R, our prior-integration mechanism is distinctly different from Pow3R.
>
>
> 3. Q1: Depth type and marginal improvement
>
> We use ground truth depth throughout training and evaluation (sensor depth from Kinect for 7-Scenes, synthetic for NRGBD, depth from LiDAR scanners for Waymo and ScanNet++).
>
>
> | Dataset| Metric    | no D  |   D   | Δ,%|
> |--------|-----------|-------|-------|----|
> | 7-Scenes| acc mean  | 0.333 | 0.313 | 4  |
> |        | acc med   | 0.262 | 0.250 | 6  |
> |        | comp mean | 0.207 | 0.195 | 6  |
> |        | comp med  | 0.171 | 0.166 | 3  |
> |        | nc mean   | 0.663 | 0.690 | 4  |
> |        | nc med    | 0.742 | 0.778 | 5  |
> | NRGBD  | acc mean  | 0.246 | 0.249 | -1 |
> |        | acc med   | 0.145 | 0.089 | 39 |
> |        | comp mean | 0.195 | 0.148 | 24 |
> |        | comp med  | 0.097 | 0.055 | 43 |
> |        | nc mean   | 0.708 | 0.746 | 5  |
> |        | nc med    | 0.829 | 0.889 | 7  |
>
> | Dataset | Metric    | no D  |   D   | Δ,%|
> |---------|-----------|-------|-------|----|
> | Waymo   | L2/1      | 1.235 | 1.042 | 16 |
> |         | L2/2      | 1.259 | 1.095 | 13 |
> |         | L2/3      | 1.300 | 1.145 | 12 |
> |         | L2/4      | 1.327 | 1.181 | 11 |
> |ScanNet++| L2/1      | 0.049 | 0.042 | 14 |
> |         | L2/2      | 0.074 | 0.060 | 19 |
> |         | L2/3      | 0.075 | 0.061 | 19 |
> |         | L2/4      | 0.086 | 0.063 | 27 |
>
>
>
>
> Overall, adding depth yields consistent improvements across all evaluated datasets. On 7-Scenes (acc/comp/nc metrics), the relative improvement is below 10%. On NRGBD (same metrics), the improvement is strong, reaching up to 43%. On Waymo (L2 pointmap error), depth reduces error by 11–16%, while on ScanNet++ it provides 14–27% relative improvement.

---

> ### Author Response · Authors · 2025-11-21
> **Response to the questions [Part 2/2]**
>
> 4. Q2: Shared vs separate encoders
>
> We have conducted two additional experiments (training and testing on the combined ScanNet++ and Waymo split because of time constraints):
> * one version with a shared VIT encoder for all guidance modalities
> * one version with separate modality-specific encoders.
> Results show that the shared encoder performs very well on ScanNet++ but is clearly outperformed by separate encoders on the more challenging Waymo dataset (outdoor scenes, large scale, dynamic objects).  On easier / indoor scenes the shared encoder remains competitive.  This supports our design choice of separate encoders for the full method while indicating that a shared encoder can be a reasonable lighter alternative in simpler settings.
>
>
>
> Metrics for ScanNet++ dataset
>
> |  Encoder | K | R,t | D | Acc Mean | Acc Med | Comp Mean | Comp Med | NC Mean | NC Med |
> |:--------:|---|-----|---|:--------:|:-------:|:---------:|:--------:|:-------:|:------:|
> | Separate | - | -   | - |    0.142 |   0.104 |      0.09 |    0.052 |   0.707 |  0.805 |
> |  | + | -   | - |    0.143 |   0.107 |     0.092 |    0.055 |   0.706 |  0.803 |
> |  | - | +   | - |    0.136 |   0.099 |     0.087 |     0.05 |   0.704 |    0.8 |
> |  | - | -   | + |     0.13 |    0.09 |     0.075 |    0.037 |   0.717 |  0.816 |
> |  | + | +   | - |    0.128 |   0.091 |     0.082 |    0.044 |   0.705 |  0.802 |
> |  | + | -   | + |    0.129 |   0.094 |     0.077 |    0.041 |   0.715 |  0.812 |
> |  | - | +   | + |    0.121 |   0.079 |     0.081 |    0.039 |   0.722 |  0.825 |
> |  | + | +   | + |    0.113 |   0.072 |     0.077 |    0.031 |   0.723 |  0.826 |
> |  Shared  | - | -   | - |    0.146 |   0.105 |      0.09 |    0.056 |   0.708 |   0.81 |
> |    | + | -   | - |    0.147 |   0.105 |      0.09 |    0.055 |   0.708 |   0.81 |
> |    | - | +   | - |    0.133 |   0.089 |     0.091 |    0.047 |    0.71 |  0.814 |
> |    | - | -   | + |    0.132 |   0.098 |     0.077 |    0.046 |   0.711 |  0.811 |
> |    | + | +   | - |     0.13 |   0.088 |     0.092 |    0.045 |   0.711 |  0.814 |
> |    | + | -   | + |    0.133 |   0.099 |     0.077 |    0.047 |   0.713 |  0.813 |
> |    | - | +   | + |    0.123 |   0.078 |      0.08 |    0.036 |   0.719 |  0.822 |
> |    | + | +   | + |    0.118 |   0.077 |     0.081 |    0.035 |    0.72 |  0.824 |
>
>
> Metrics for Waymo dataset
>
> |  Encoder | K | R,t | D | Acc Mean | Acc Med | Comp Mean | Comp Med | NC Mean | NC Med |
> |:--------:|---|-----|---|:--------:|:-------:|:---------:|:--------:|:-------:|:------:|
> | Separate | - | -   | - |    0.753 |   0.482 |     1.154 |    0.463 |   0.709 |  0.822 |
> |  | + | -   | - |    0.754 |   0.476 |     1.152 |    0.461 |   0.709 |  0.822 |
> |  | - | +   | - |    0.704 |   0.456 |     1.095 |    0.431 |   0.704 |  0.817 |
> |  | - | -   | + |    0.709 |   0.472 |      0.87 |    0.416 |   0.712 |  0.825 |
> |  | + | +   | - |    0.712 |   0.466 |     1.103 |    0.443 |   0.704 |  0.816 |
> |  | + | -   | + |    0.732 |   0.501 |     0.891 |    0.444 |   0.711 |  0.825 |
> |  | - | +   | + |    0.701 |   0.462 |     0.847 |    0.403 |    0.71 |  0.825 |
> |  | + | +   | + |    0.682 |   0.444 |     0.832 |    0.389 |   0.711 |  0.825 |
> |  Shared  | - | -   | - |     0.76 |   0.508 |      1.21 |    0.485 |   0.705 |  0.818 |
> |    | + | -   | - |    0.757 |   0.504 |     1.207 |     0.48 |   0.705 |  0.818 |
> |    | - | +   | - |    0.725 |   0.456 |     1.135 |    0.441 |   0.707 |  0.818 |
> |    | - | -   | + |    0.831 |   0.602 |     1.004 |    0.553 |   0.706 |  0.821 |
> |    | + | +   | - |    0.725 |   0.459 |     1.141 |    0.446 |   0.708 |  0.819 |
> |    | + | -   | + |    0.796 |   0.569 |     0.976 |    0.522 |   0.707 |  0.822 |
> |    | - | +   | + |    0.711 |   0.465 |     0.874 |    0.406 |   0.709 |  0.822 |
> |    | + | +   | + |    0.736 |   0.498 |       0.9 |    0.447 |   0.708 |  0.821 |
>
>
>
> 5. Q3: Pow3R in Tab. 3
>
> The Pow3R results reported in Tab. 3 follow the original Pow3R design and training protocol, without any of the prior-integration mechanisms proposed in our work.

---

> > ### Comment · Reviewer_J8qk · 2025-11-26
> >
> > Thank authors for the responses. I have several follow-up questions:
> >
> > - What is the number of parameters in CUT3R and G-CUT3R?
> > - In *Answer 2*, how many hours are required to train G-CUT3R? Could the authors provide concrete numbers to support the claim that “our method requires less training time/data”? Specifically, how much training data is used, and how many training hours are needed for Pow3D and CUT3R? In addition, since this work freezes the encoder, it may converge faster but potentially degrade overall performance.
> > - In *Answer 4*, the first-row results in the tables for the “separate model” and “shared model” differ. Is this correct? These settings do not apply any prior and should correspond to the baseline. Shouldn’t these results come from the same model?

---

> > > ### Author Response · Authors · 2025-11-27
> > >
> > > Thank you very much for your response and additional questions.
> > >
> > > Q1: What is the number of parameters in CUT3R and G-CUT3R?
> > >
> > > Here is calculated number of parameters for CUT3R and G-CUT3R models:
> > >
> > > CUT3R: 463,438,546
> > >
> > > G-CUT3R: 532,819,599 (5% overhead per modality, 15% including all modalities)
> > >
> > > Q2: How many hours was G-CUT3R trained? How much training data was used? Does freezing the encoder help?
> > >
> > > Hours: We have trained our model for approximately 15 epochs on 4xA100, one epoch takes ~15 hours.
> > >
> > > Amount of data: We have used 12 datasets (whereas CUT3R was trained on 33 datasets) as we report in Tab. 4 in the supplementary material.
> > >
> > > Does freezing help: We do not report the ablation on the frozen / trained encoder, but in our earlier experiments, we did not see any improvement from training the encoder. Moreover, training the encoder instead of freezing it sometimes made results even worse. We think this happens because we add our prior only to the decoder, and training the encoder on a smaller subset of data spoils the performance.
> > >
> > > Q3: Why shared vs separate encoder results differ.
> > >
> > > We have trained two separate models from scratch, which explains the differences in results.. In the first row referenced by the reviewer, no extra information is provided, and both models receive exactly the same input. Since the difference is small, we believe the difference in the models can be explained by varying results due to different random seeds.

---

### Author Response · Authors · 2025-11-22
**Comment**

We deeply appreciate the efforts of all reviewers! We have revised the manuscript and replied to all reviewers individually. We are looking forward to your responses. Thank you so much again.

---

### Author Response · Authors · 2025-12-02
**Summary for AC**

We want to thank all reviewers for their effort and contribution to our work. Thanks to our short, but insightful discussion we were able to revise our manuscript and answer all questions.
Here we have prepared a short summary of our discussions:

**1. Efficiency and lightweight claims**

We report FPS and number of parameters for our model and G-CUT3R.
Parameters:
CUT3R: 463,438,546
G-CUT3R: 532,819,599 (5% overhead per modality, 15% including all modalities)
FPS:
CUT3R: 20
G-CUT3R: 14-18

**2. Differences from Pow3r and CUT3R**

Key differences are the following:
* We better preserve the capabilities of the prior model thanks to zero-initialized convolutions during fusion. This design(as shown in our ablation) enables faster convergence and training on only a subset of CUT3R’s data.
* We introduce lightweight, modality-specific VIT encoders that project each prior into a suitable fusion space, rather than relying on simple addition.
* Our design also allows fusing directly to the decoder, allowing us to keep the whole encoder frozen. Pow3r needs to retrain the encoder and decoder.
Thus, while we build on CUT3R, our prior-integration is different from Pow3R and is faster and has better metrics.

**3. Depth type and marginal improvement**

Overall, adding depth yields consistent improvements across all evaluated datasets. On 7-Scenes (acc/comp/nc metrics), the relative improvement is below 10%. On NRGBD (same metrics), the improvement is strong, reaching up to 43%. On Waymo (L2 pointmap error), depth reduces error by 11–16%, while on ScanNet++ it provides 14–27% relative improvement.

**4. Noise robustness analysis**

We have evaluated robustness by injecting Gaussian noise directly into the additional prior modalities. Noise has been sampled from a normal distribution with mean = 0 and std ranging from 5% to 50% of the ground truth data.)
The new results show that G-CUT3R consistently outperforms the non-guided baseline even at a noise level of up to 20%, and it degrades more significantly at higher noise. This confirms the robustness of our guided fusion mechanism to realistic sensor noise.
Results are reported in the supplementary material **APPENDIX E**.

Thank you again for our discussion. We believe we have addressed all questions and concerns raised by reviewers. In our manuscript, we provide a comprehensive comparison with all competing methods in Tab. 1, where our proposed G-CUT3R achieves the best reconstruction accuracy while being real-time and highly robust to varying noise levels in the prior data.

---

### Meta-Review · Area_Chair_WfgJ · 2026-01-06

**Summary:**

G-CUT3R extends CUT3R for feed-forward multi-view 3D reconstruction by fusing optional priors (depth, intrinsics, poses) via modality-specific encoders and zero-initialized convolution layers in the decoder. A single model is trained to support arbitrary combinations of priors using random modality subsets. Across 7-Scenes, NRGBD, and related tasks, the paper reports consistent gains over CUT3R and DUSt3R-family baselines, along with a large speed advantage over Pow3R. Reviewers, however, question the limited conceptual novelty and note that the paper does not clearly report model-size and compute overhead.

**Reviewer Concerns:**

The rebuttal clarifies depth and pose sources by dataset, adds noise sensitivity results for priors, and provides parameter counts and training setup details. However, concerns remain about limited conceptual novelty and whether the added modules materially increase compute and memory costs in common settings beyond the reported FPS, especially across different modality combinations.

**Reviewer Scores:**

Reviewer J8qk: likely 4->5 after seeing parameter counts, training details, and clearer depth/pose explanations.

Reviewer bCZM: likely 4->5 with the added dataset-by-dataset depth/pose table and noise robustness plots.

Reviewer iLL2: would likely rescore to 3-4 after realizing the initial review discussed different components and re-evaluating the actual submission.

---

### Decision · Program_Chairs · 2026-01-26

Reject